# Wearable Devices for Biofeedback Rehabilitation: A Systematic Review and Meta-Analysis to Design Application Rules and Estimate the Effectiveness on Balance and Gait Outcomes in Neurological Diseases

**DOI:** 10.3390/s21103444

**Published:** 2021-05-15

**Authors:** Thomas Bowman, Elisa Gervasoni, Chiara Arienti, Stefano Giuseppe Lazzerini, Stefano Negrini, Simona Crea, Davide Cattaneo, Maria Chiara Carrozza

**Affiliations:** 1IRCCS Fondazione Don Carlo Gnocchi ONLUS, 20019 Milan, Italy; tbowman@dongnocchi.it (T.B.); carienti@dongnocchi.it (C.A.); slazzarini@dongnocchi.it (S.G.L.); simona.crea@santannapisa.it (S.C.); dcattaneo@dongnocchi.it (D.C.); mccarrozza@dongnocchi.it (M.C.C.); 2The Biorobotics Institute, Scuola Superiore Sant’Anna, 56121 Pisa, Italy; 3Department of Biomedical, Surgical and Dental Sciences, University “La Statale”, 20100 Milan, Italy; Stefano.Negrini@unimi.it; 4IRCCS Istituto Ortopedico Galeazzi, 20100 Milan, Italy; 5Department of Physiopathology and Transplants, University of Milan, 20100 Milan, Italy

**Keywords:** rehabilitation, inertial measurement unit, force sensors, biofeedback, postural balance, gait, stroke, Parkinson’s disease, mild cognitive impairment

## Abstract

Wearable devices are used in rehabilitation to provide biofeedback about biomechanical or physiological body parameters to improve outcomes in people with neurological diseases. This is a promising approach that influences motor learning and patients’ engagement. Nevertheless, it is not yet clear what the most commonly used sensor configurations are, and it is also not clear which biofeedback components are used for which pathology. To explore these aspects and estimate the effectiveness of wearable device biofeedback rehabilitation on balance and gait, we conducted a systematic review by electronic search on MEDLINE, PubMed, Web of Science, PEDro, and the Cochrane CENTRAL from inception to January 2020. Nineteen randomized controlled trials were included (Parkinson’s *n* = 6; stroke *n* = 13; mild cognitive impairment *n* = 1). Wearable devices mostly provided real-time biofeedback during exercise, using biomechanical sensors and a positive reinforcement feedback strategy through auditory or visual modes. Some notable points that could be improved were identified in the included studies; these were helpful in providing practical design rules to maximize the prospective of wearable device biofeedback rehabilitation. Due to the current quality of the literature, it was not possible to achieve firm conclusions about the effectiveness of wearable device biofeedback rehabilitation. However, wearable device biofeedback rehabilitation seems to provide positive effects on dynamic balance and gait for PwND, but higher-quality RCTs with larger sample sizes are needed for stronger conclusions.

## 1. Introduction

People with neurological diseases (PwND) show mobility disorders including balance and gait deficits leading to a slowing of gait speed and an increased risk of fall [1,2,3] which consequently impact the quality of life [4,5,6].

Gait and balance characteristics of patients with Parkinson’s disease (PD) or following a stroke have been largely documented. In patients with PD, the progressive loss of dopamine in the basal ganglia can lead to gait patterns such as festination, shuffling steps, and freezing of gait [7]. Poststroke hemiplegic gait is characterized by asymmetry and reduced weight-bearing on the paretic limb, as a result of the residual functions after upper motor neuron syndrome [8]. These patients’ populations have specific rehabilitation needs: for example, PD rehabilitation programs typically focus on increasing the step length, whereas stroke rehabilitation mostly aims to improve weight-bearing on the affected leg.

Over the years, rehabilitation has had a key role in the treatment of these impairments for PwND [9], and several studies have demonstrated the positive effects of rehabilitation in improving gait and balance disorders [10,11,12]. Between different rehabilitation approaches, biofeedback is a well-established technique used in rehabilitation settings to facilitate normal movement patterns after injury [13,14]. 

Biofeedback’s approach is based on well-established theories of motor learning [15], providing augmented information, or “signal” on biomechanical or physiological parameters that are obtained by measuring body movement and force, cardiovascular, or neurological parameters [16]. A biofeedback signal is based on four main components: mode, content, frequency, and timing [17,18]. Mode refers to the channel used to provide information to the user and can be visual, auditory, vibrotactile, or a combination. Content refers to the information provided to the user and can be grounded on performance (e.g., execution of the movement) or results (e.g., outcome of the movement) [19]. Frequency refers to the number of signal event occurrences per unit of time and can be constant, reduced, or fading throughout the rehabilitation process. Timing refers to when the signal is given with respect to movement execution. Timing can be concurrent, if the feedback is delivered during the execution of the movement, or terminal, if delivered at the end of the movement [20,21]. Moreover, biofeedback can be used for positive or negative reinforcement: positive reinforcement aims to increase a specific movement pattern (to make it occur more frequently), whereas negative reinforcement occurs when the user is asked to reduce a certain movement pattern or behaviour (negative reinforcement aims at making an event occur less frequently) [22,23].

Based on previous components, biofeedback can be applied by clinicians using the principles of motor learning to improve awareness of exercise and to facilitate normal movement patterns in PwND [13,24,25].

In this context, the emerging wearable devices can provide a biofeedback signal to maximize patients’ improvements. Wearable devices have already had an impact on healthcare practices by allowing continuous monitoring of human parameters inside and outside the clinic in different health areas (e.g., cardiovascular, gastrointestinal, sleep, neurology, and physical activity) [26]. In addition, the portability and ease of use of wearable devices make them suitable for clinical rehabilitation and home rehabilitation to improve rehabilitation effectiveness [27]. Indeed, wearable devices incorporated sensors, measuring biomechanical or physiological variables of patients as required by biofeedback rehabilitation, with the advantage of being portable in any rehabilitation setting [28]. For example, sensors can be attached to shoes, canes, clothing, gloves, and watches or can be skin attachable.

Different types of sensors are integrated as inertial measurement units (IMU), incorporating accelerometers, gyroscopes, and magnetometers able to provide biofeedback about the estimated dynamics of the body centre of mass or upper- and lower-extremity movement [29]. Similarly, pressure sensors can be used for measuring plantar force during standing and walking tasks to estimate the user’s posture and weight-bearing, while electroencephalographic (EEG) or electromyographic (EMG) sensors measure brain or muscles signals [30,31,32].

Based on the abovementioned sensory and biofeedback components, wearable devices can be used with PwND in wearable devices-based biofeedback rehabilitation (WDBR) [33].

Nevertheless, it is not yet clear what the most commonly used or practical configurations (e.g., number and placement) are in terms of sensors and biofeedback components depending on the specific deficit/pathology. Moreover, training paradigms using wearable devices should be analysed to verify if motor-learning principles are properly applied in published rehabilitation studies, and design rules should be provided to develop effective and user-friendly wearable devices facilitating their adoption in everyday clinical practice. Finally, a previous review of randomized controlled trials (RCT) [34] showed a positive effect of WDBR on gait and balance outcomes for older adults and patients. Although these results were promising, a greater number of RCTs were needed and have been included in this review as the most robust study design to estimate the feasibility and effectiveness of WDBR in clinical practice [34].

Thus, the objective of this systematic review of RCTs is twofold: First, to analyse the state-of-the-art technology on WDBR and present an overview of wearable devices in terms of their sensor configurations, biofeedback components, and training paradigms to provide practical design rules. Second, to evaluate the feasibility, usability, and estimated effectiveness of WDBR on balance and gait outcomes in PwND.

## 2. Materials and Methods

### 2.1. Data Sources and Searches

The review protocol was registered in PROSPERO (CRD42020162957) and reported according to the PRISMA statement [35]. The following databases were searched: MEDLINE, PubMed, Web of Science, PEDro, and the Cochrane Central Register of Controlled Trials. Additional relevant papers were found by hand in trial registers or other grey literature sources. Studies published in the English language from inception to January 2020 were included. Relevant search terms were combined with boolean operators (OR/AND) as reported in Table 1 and Appendix A.

### 2.2. Eligibility Criteria

Eligibility criteria according to the PICOs were:

(P) Participants: adults with any neurological diseases or disorders;

(I) Intervention: wearable devices biofeedback rehabilitation for balance or gait;

(C) Comparators: balance and gait rehabilitation without wearable devices or biofeedback, conventional rehabilitation, or usual care, no training;

(O) Outcomes: balance and gait;

(s) Study design: randomized controlled trials (RCTs).

To have been included in the systematic review, studies had to be focused on adults (age >18 years) with any type of central neurological disease or disorder. Moreover, studies should have focused on rehabilitation intervention using wearable devices and biofeedback principles. In addition, only studies that evaluated walking, ambulation, postural balance, equilibrium, motor activity, and recovery of function using spatiotemporal gait parameters, instrumental indexes, and clinical scales have been included. 

Studies were excluded from this review if they did not use wearable devices or did not provide biofeedback. Moreover, studies not written in English, with relevant missing information, or that were out of topic with respect to the aims of the present study were excluded. Finally, we excluded all study designs except for RCTs. 

### 2.3. Study Selection and Data Extraction

The studies retrieved using the search strategy and additional sources were screened independently by two main reviewers (TB and EG), based on their titles and abstracts, and considering inclusion and exclusion criteria. Then, the full text of eligible studies was further analysed and independently assessed for eligibility. Any disagreement between the two reviewers over the eligibility of studies was resolved through discussion with a third reviewer (DC).

After inclusion, the study characteristics, research goals, and main findings were extracted and summarized. Extracted information also included: study setting, study population, participant characteristics, wearable device, biofeedback characteristics, details of the intervention and control conditions, study methodology, outcomes and times of measurement, indicators of acceptability, and feasibility.

### 2.4. Data Synthesis

To evaluate the effects of WDBR on balance and gait outcomes, we merged in a meta-analysis study with common characteristics of intervention. Firstly, we considered if wearable devices had been used as “add-on” therapy, or “not add-on”, in the experimental group compared to the control group. In studies where devices had been applied as “add-on”, both groups performed the same exercises for balance and gait with the only difference being the addition of the device providing biofeedback information in the experimental group. In all other cases, wearable devices have been classified as “not add-on”. Secondly, studies have been grouped according to the key components of balance rehabilitation exercises proposed by Horak et al. [36]. The following 6 components were considered: (1) biomechanical constraints, defined as the size, quality of the base of support and any limitations in strength, range of motion, and pain of the feet; (2) movement strategies, defined as the strategies used to return the body to equilibrium in a standing position; (3) sensory strategies, defined as the sensory information from somatosensory, visual, and vestibular systems that are integrated to interpret complex sensory environment; (4) orientation in space, defined as the ability to orient the body parts with respect to gravity; (5) control of dynamics (gait and proactive balance), defined as the control of balance during gait and while changing from one posture to another; and (6) cognitive processing, defined as the cognitive resources required during exercises and postural control.

Studies using the wearable devices as “add-on” therapy and exercises with similar rehabilitation components were pooled into the meta-analysis. First, we determined the overall effect of WDBR versus control intervention on different balance and gait outcomes at postassessment and follow-up (FU) in PwND. Second, to assess the effectiveness of different wearable devices on specific patient’s populations, a subgroup analysis has been performed. Studies were grouped according to the wearable sensor type (e.g., IMU sensors and pressure sensors) and the neurological disease (e.g., stroke and Parkinson’s).

All meta-analyses have been performed using random effects model and calculating the mean differences (MD) and 95% of confidence interval (CI) to acknowledge the methodological and clinical differences among studies [37]. Heterogeneity of the studies was assessed using the inconsistency test (I^2^), whose values could be interpreted as follows: from 0% to 40% low heterogeneity; from 30% to 60% may represent moderate heterogeneity; from 50% to 90% may represent substantial heterogeneity; and from 75% to 100%: considerable heterogeneity [38]. 

Meta-analyses were calculated using Review Manager 5.3 (the Nordic Cochrane Centre, Copenaghen, Denmark). Alpha level was set at 0.05 to test for overall effect.

All studies not included in the meta-analysis have been considered in a qualitative synthesis of the results (see Appendix A).

### 2.5. Risk of Bias Assessment

The risk of bias for all included RCTs was assessed with the six domains defined by the Cochrane Collaboration’s tool [39]. These six domains are: (1) selection bias, due to random sequence generation and allocation concealment; (2) performance bias, with blinding of participants and personnel as a possible source of bias; (3) detection bias, due to blinding of outcome assessment; (4) attrition bias, evaluating possible incomplete outcome data; (5) reporting bias, due to selective outcome reporting; and (6) other bias, evaluating any important concerns about bias not covered in the other domains. Each domain was judged as “low risk of bias” (“green”), “high risk of bias” (“red”), or “unclear risk of bias” (“yellow”).

## 3. Results

After a database search, 8065 potentially relevant papers were found (Figure 1). After removing duplicates, 4224 article titles and abstracts were screened for relevance. Nineteen papers were finally included. 

### 3.1. Study Characteristics

Table 2 shows participant characteristics, wearable devices, type of sensors and placement, experimental and control intervention, timing of the intervention, balance and gait outcomes, and finally time points of the included studies. The selected RCTs were performed in people with PD (n = 6) [40,41,42,43,44,45], stroke (n = 13) [41,46,47,48,49,50,51,52,53,54,55,56,57] and mild cognitive impairment (MCI) (n = 1) [58]. The total sample consists of 513 PwND with mean age ranging from 46 to 79 years. In each study, samples size ranged from a minimum of 10 to a maximum of 42 subjects included.

### 3.2. Wearable Device Biofeedback Rehabilitation: Sensors Classification and Configuration

Based on the classification by Giggins et al. [14], we found 13 studies providing biomechanical measurements and six studies providing physiological measurements of the body system, using wearable devices based on different type of sensors (Figure 2).

Considering biomechanical measurements, five studies [43,46,48,52,53] used pressure sensors and one study [57] combined a pressure sensor with a foot switch. Pressure sensors were positioned under the foot of the paretic leg and were activated considering the % body weight loading [53,57] or by the combination of body weight loading and specific gait cycle phase (Figure 3) [46,48]. In Jung et al. [52], pressure sensors were embedded into a cane used on the nonparetic side and activated by body weight %. Only El Tamawy et al. [43] used pressure sensors in both shoes to detect the push-off phase during gait.

Conversely, only three studies [42,44,58] used IMU sensors alone, while Byl et al. [41] combined IMU sensors with wearable pressure sensors, and the other two studies [45,55] combined IMU sensors with a force platform. IMU sensors were always positioned in the lower limbs, with different configurations involving thighs, shanks, and shoes. In the study by Carpinella et al. [42] IMU sensors were positioned on the upper trunk and the lower trunk, as well as on the lower limbs. Similarly, Schwenk et al. [58] placed IMU sensors on the lower trunk and on the lower limbs. In Van der Heuvel et al. [45], IMU sensors were positioned on the trunk and combined with the force platform, while Lupo et al. [55] placed sensors on the trunk, midthigh level and at midtibial level of the affected or the healthy side, depending on the exercise. Finally, only Cho et al. [47] used an electrogoniometer in the lower limb to measure the knee angle.

Considering physiological measurements, two studies [40,54] used EEG sensors and four studies [49,50,51,56] used EMG sensors. In both studies using EEG, electrodes were placed in compliance with the International 10–20 System. Lee et al. [54] secured an electrode to the scalp over the region of the central lobe (Cz), while Azerpaikan et al. [40] attached two electrodes to the left and right occipital (O1, O2) and one to the subject’s left earlobe. 

EMG signals were recorded with surface electrodes placed over the tibialis anterior muscle belly of the affected leg [49,50] or over the gastrocnemius lateralis muscles according to Seniam guidelines [49,51]. In one study, EMG sensors recorded the gastrocnemius and pretibial muscles activity in combination with an electrogoniometer inserted in the subjects’ shoes to measure ankle movements [56].

### 3.3. Modalities of Exercise Interventions

The duration of the whole rehabilitation period ranged from a minimum of 10 sessions to a maximum of 30 sessions over 2 months. Single session dedicated time varied from 20 to 90 mins of training; in only two studies, treatment’s time was not specified [53,56]. With respect to treatment frequency, the majority of the studies reported a frequency ranging from 2/week to 5/week, and in only two studies information was not reported [41,53].

Characteristics of the balance and gait training components of the biofeedback signal were different depending on the studies.

### 3.4. Components of Balance and Gait Training

According to the framework of Horak et al. [36], most of the studies (16 out of 19) involved the control of dynamics during stepping, body weight shift, changing from one posture to another, and walking exercises. Sensory strategies (e.g., balance activities on stable, unstable, and moving surfaces with eyes closed and eyes open) have been practiced in two studies [41,42]. Movement strategies (e.g., practicing balance with feet together) were applied only in the study by Byl et al. [41]. Two studies [41,43] took into account exercises aimed at improving orientation in space by standing in an upright position in front of a mirror to maintain a good postural alignment; moreover, eight studies [41,43,45,47,49,50,56,57] spent treatment’s time to improve biomechanical constraints (e.g., increasing muscles strength and training limits of stability). Noteworthy, all the studies involved cognitive processing since cognitive resources are essential during exercises and for the postural control required for balance and gait training and to elaborate information provided by the feedback itself [59]. 

Treatment progression was described in 6 out of 19 studies, and in some of these studies, the physiotherapist progressively adjusted training complexity by changing the reference values of the task or exercise, according to the ability of each patient [41,42,45,51]. In the study by El-Tamawy et al. [43], the treadmill walking time was increased gradually from 6 to 25 mins, and walking speed progression was self-selected by each subject, while Intiso et al. [50] described two progressive phases of the training increasing the needed threshold to provide biofeedback when the patient made errors of less than 20% during the session.

### 3.5. Biofeedback Components

In Table 3, we reported biofeedback components as mode, content, frequency, and timing. Moreover, we have considered if the authors provided any explanation about the progression of training in line with motor-learning principles. Regarding timing, all the devices but one [41] provided concurrent, or ‘real-time’, biofeedback. On the other hand, Byl et al. [41] provided terminal biofeedback while the subject was standing or sitting after performing a few walking trials. In the same way in Carpinella et al. [42], a terminal feedback rating performance was given to the patient after each exercise. 

Biofeedback content was related to knowledge of the performance (e.g., information about movement coordination or muscle activity during the movement) in all the studies included. In addition, studies [41,42,45,55] also provided knowledge of results, presenting a scoring point or a number representing the outcome of the performance. In eight of the studies [44,46,48,50,51,52,53,57], the modality exploited to transmit the signal to users’ devices was auditory. Four studies [40,41,45,47,54] used visual signal, while in the other five studies [42,49,55,56,58] a combination of auditory and visual biofeedback was applied. Only El-Tamawy et al. [43] provided vibrotactile biofeedback.

According to principles of motor learning, a fading frequency of the biofeedback signal was provided in two studies [42,44]. Jonsdottir et al. [51] provided a constant signal in the first phase of the training and a fading signal in the second phase. The remaining 17 studies provided a constant frequency (e.g., signal provided every time a biomechanical or physiological variable reached a predefined threshold). 

Considering the type of reinforcement, in 12 out of 19 studies, a positive reinforcement signal was given when the variable measured by the device remained within a pre-established therapeutic window or reached a predetermined threshold [40,43,45,46,48,49,50,51,53,54,55,56,57]. Instead, in the study by Jung et al. [52], negative reinforcement has been provided since stroke patients were instructed to avoid activating the beeping sound from the cane with the aim of increasing load in the paretic limb. In four studies [41,42,44,58], biofeedback signal was positive or negative depending on the task or the activity performed, and in two studies [47,55], information about the type of reinforcement was not specified.

### 3.6. WDBR Estimated Effectiveness on Balance and Gait Outcomes

#### 3.6.1. Berg Balance Scale (BBS)

Three studies [42,55,57] compared the effects of add-on WDBR training on BBS with controls. Our meta-analysis revealed a significant overall effect in favour of add-on WDBR at postassessment (MD = 4.99 [1.79, 8.18]; *p* = 0.002; Figure 4). The same overall effect was maintained in studies with FU assessment (MD = 5.29 [0.06, 10.51]; *p* = 0.05; Figure 4). Homogeneity criteria were met for all the analyses (I^2^ = 0%; Chi^2^ = 0.30, *p* = 0.84). A qualitative synthesis for studies not included in the meta-analysis (see Appendix A) showed no significant differences between groups in two studies [41,45], instead Azerpaikan et al. [40] found significant differences (*p* < 0.01) in favour of neurofeedback provided by EEG device.

#### 3.6.2. Timed up and Go (TUG)

Five studies [42,46,48,53,57] compared the effects of add-on WDBR training on TUG with controls. Our meta-analysis revealed a significant overall effect in favour of add-on WDBR at postassessment (MD = −3.43 s [−6.53, −0.32]; *p* = 0.03; Figure 5). Homogeneity criteria were met (I^2^ = 7%; Chi^2^ = 4.28, *p* = 0.37). Among studies not included in the meta-analysis (qualitative synthesis) (see Appendix A), Byl et al. [41] found no significant differences between groups.

Subgroup analysis, considering only devices embedded with pressure sensors and providing auditory feedback, revealed no significant effect in favour of add-on WDBR at postassessment in stroke population (MD = −2.31 s [−5.44, 0.82]; *p* = 0.15; Figure 5). Homogeneity criteria were met (I^2^ = 0%; Chi^2^ = 0.94, *p* = 0.81). 

#### 3.6.3. Gait Speed

Six studies [42,44,46,48,52,57] compared the effects of add-on WDBR training on gait speed compared to controls. A meta-analysis of these studies revealed no significant overall effect at postassessment (MD = 0.08 m/s [−0.00, 0.17]; *p* = 0.06; Figure 6) and at FU assessment (MD = 0.13 m/s [−0.18, 0.44]; *p* = 0.42; Figure 6). Subgroup analysis [46,48,52,57] considering only device embedded with pressure sensors and providing auditory feedback, revealed significant effect in favour of add-on WDBR at postassessment (MD = 0.08 m/s [0.01, 0.14]; *p* = 0.91; Figure 6). Homogeneity criteria were met (I^2^ = 0%; Chi^2^ = 0.55, *p* = 0.81). 

Three studies [41,45,47] included in qualitative synthesis (see Appendix A) found no differences comparing add-on WDBR to control. Similarly, Intiso et al. [50] found no differences between groups, providing EMG-based biofeedback in addition to standard physical therapy compared to standard physical therapy alone. 

Jonsdottir et al. [51] provided task-oriented gait training with EMG, while Cozean et al. [49] performed EMG biofeedback combined with FES during static and dynamic activities. Both studies showed significant improvements (*p* = 0.04) in favour of the experimental group compared to the conventional physical therapy group, and results were maintained at FU assessment (*p* = 0.02). Similarly, Mandel et al. [56] found significant differences (*p* = 0.04) in the experimental group (EMG biofeedback followed by rhythmic positional biofeedback) compared to no treatment, and results were maintained at FU assessment (*p* = 0.035). Finally, El Tamawy et al. [43] found significant differences (*p* = 0.001) in favour of a treadmill with biofeedback group compared to control, and Lee et al. [54] found significant differences (*p* = 0.05) in favour of Neurofeedback group compared to sham therapy.

#### 3.6.4. Qualitative Synthesis

Due to heterogeneity between studies, the remaining outcome measures were not pooled into the meta-analysis but have been considered in a qualitative synthesis of the results (see Appendix A).

### 3.7. Feasibility and Usability of WS Training

Only four studies [42,44,45,58] evaluated the feasibility and usability of wearable devices. In the study by Carpinella et al. [42], the Tele-healthcare Satisfaction Questionnaire-Wearable Technology showed that all patients, but one found the wearable device beneficial. Moreover, it was considered reliable, safe, and easy to use by all the patients, and comfortable by 15 out of 17 subjects. Among all, 65% found that using the wearable devices required effort and that such effort was worthwhile for them. Physiotherapists appreciated the wearable devices, but they suggested reducing the number of sensors and to simplify calibration procedures. In the study by Ginis et al. [44], participants were very positive about the system, and scores on user-friendliness were on average above 4 on a 5-point Likert scale. Further, it was observed that participants with previous smartphone experience had the least problems using the system. In the study by Schenk et al. [58], participants described their experience using the technology with an adapted questionnaire. Most participants stated that it was fun. Likewise, most participants rated the usage, form, and design of the technology positively. They felt safe while using it, never experiencing fear of falling, and without the need for balance support during the therapy. For most participants, the balance exercises were not difficult to perform and were not too fast. Finally, Van Der Hovel et al. [45] did not use outcome measures to assess patients’ perspective, but feasibility and usability were reported by the therapists involved in the training. They confirmed that the device-based therapy was well accepted by most participants, with the element of scoring being appreciated. They also observed that less-disable patients could operate the workstations independently, conversely patients with higher disability required more assistance. Furthermore, the system was considered suitable for use in a group setting where continuous one-on-one supervision is not needed. 

### 3.8. Risk of Bias (RoB) Assessment 

Risk of bias graph is reported in Figure 7. Most of the studies (more than 75%) present unclear risk of bias in allocation concealment (selection bias), blinding of participants and personnel (performance bias), blinding of outcome assessment (detection bias), incomplete outcome data (attrition bias), and selective reporting (reporting bias). Considering “other bias”, all the studies present unclear risk of bias.

The risk of bias summary (Figure 8) reveals that only 8 out of 19 studies presented a low risk of bias and four [41,43,46,48] high risk of bias in “random sequence generation”. Conversely, 3 of 19 studies showed high risk of bias in “blinding of outcome assessments”. Two studies [41,43] had high risk of bias in “blinding of participants and personnel”, and one study [41] had high risk of bias in all the remaining bias.

## 4. Discussion

To the best of our knowledge, our systematic review with meta-analysis of randomized controlled trials represents the most comprehensive synthesis to date on the type and configuration of wearable devices for rehabilitation purposes. We also considered their feasibility, usability in a clinical setting, and attempted to estimate the effectiveness of biofeedback rehabilitation using wearable devices on balance and gait outcomes in PwND.

### 4.1. Training Paradigm: Type and Configuration of Sensors

This review identified a great variety in terms of sensor type and configuration and biofeedback components used in the rehabilitation of PwND.

A direct comparison between sensor configurations is difficult due to the different methodologies and lack of information. Nevertheless, we described the most frequent paradigm used to apply biomechanical and physiological sensors for different conditions or functional disorders. Most of the studies [41,42,43,44,45,46,48,52,53,55,57,58] reported the use of biomechanical sensors as pressure and inertial sensors. As previously stated, pressure sensors were prevalently placed under patients’ feet to measure the ground reaction force generated by the body and were used to give biofeedback about weight-bearing or centre of pressure (COP) position during gait cycle phases.

The use of pressure sensors combined with auditory biofeedback has primarily been used in poststroke rehabilitation, to increase weight-bearing on the paretic leg, with promising results on gait speed improvement, as reported in the section about training effectiveness. In this review, only 2 studies out of 19 applied pressure sensors in PD. Even if PD patients present specific alterations of gait parameters easily detected and trained providing biofeedback with pressure sensors, References [60,61] further RCTs should explore this training paradigm on PD. On the other hand, IMU sensors provide sensitive measures of postural sway [14]. They can be used to estimate three-dimensional information of a body segment, such as orientation, velocity, and gravitational force, and to identify COM movement during balance training. They were positioned on the trunk and lower limbs (thighs, shanks, and feet) with different configurations depending on the study. It is noteworthy that for balance and gait training purpose, 5 out of 7 studies using IMU placed the sensors on the great mass body part (e.g., chest or lower back; Figure 3), often combined with IMU positioned on the smaller part of the lower limb (e.g., tight or shank; Figure 3).

Training paradigm using IMU has been applied in most of the studies involving PD patients. Even if further evidence on the effectiveness of this subgroup of patients should be provided, IMUs can assess specific postural problems typical of PD population. In PD subjects, postural deficits are easily measured using wearables devices to control axial segments (trunk and pelvis) and the relative position of the limbs. [42]

In this review, physiological sensors mostly involved the measurement of EMG and EEG signals. EMG sensors (surface electrodes) were mainly placed over the distal muscle (tibialis anterior or gastrocnemius) belly of the affected leg of stroke subjects. Based on our results, training paradigms using EMG sensors are mainly used in this population; in fact, no studies using EMG on PD or other neurological populations have been included. Two studies [50,56] used EMG configuration to increase the production of voluntary dorsiflexion through activation of the anterior tibialis.

Similarly, Jonsdottir et al. [51] recorded the EMG signal from the gastrocnemius lateralis muscle to provide biofeedback about performance to increase the power production of the ankle during gait. Only Cozean et al. [49] used EMG signal to increase tibialis anterior recruitment and to simultaneously induce relaxation of the gastrocnemius in those patients with moderate or severe spasticity. Only two studies [40,54] used EEG sensors as a component of a device (Procomp Infiniti system) for neurofeedback training. EEG surface electrodes were placed on the scalp in compliance with the International 10-20 System to measure brain waves. During neurofeedback training, sensorimotor rhythm wave and β wave, which are activated when focusing, were set to reward threshold; conversely, the Delta wave, which is activated when sleeping, and the Gamma wave, which is activated when nervous, were set to ‘inhibit’ threshold. 

### 4.2. Training Paradigm: Biofeedback Components

Wearable devices in the included studies mostly provided real-time biofeedback signals about performance based on positive reinforcement, with only 4 out of 19 studies providing negative reinforcement. Previous studies [22,62] suggested that positive reinforcement can be successfully used during patient’s rehabilitation resulting in greater improvement in outcomes and increased retention of the motor memory. Moreover, it has been already demonstrated that the knowledge of good performance can activate the striatum, a key region of the reward system and highly relevant for motivation [23].

In this context, the rehabilitation process can be considered as a learning environment in which real-time positive biofeedback stimulates motor learning, and wearable devices providing biofeedback should be promoted to maximize learning effects.

Only three studies [42,44,51] reported a fading progression of the biofeedback, highlighting that motor-learning principles were not properly described in most of the treatments’ protocols. Indeed, a proper progression is required to maximize the treatment’s effects phases. 

### 4.3. Feasibility and Usability of Sensor-Based Training

Only 4 studies out of 19 have provided information on the feasibility and the usability of wearable sensors. The two most often cited factors influencing the acceptability of the wearable sensors were safety and comfort, collected by the patients and therapists. Most of the studies failed to evaluate the feasibility and the usability—this is a major limitation. This means that we have few clues from the literature on strategies to reduce the effort required to use wearable devices by both therapists and patients, to reduce the time spent setting up, to regulate biofeedback threshold, and to improve data extraction. In this regard, the implementation of new technology in rehabilitation requires a meticulous approach, and different questions should be investigated (e.g., Is the device capable of reducing the workload for clinicians? Have all the factors associated with patient comfort and safety been evaluated?). Further studies that specifically focus on improving these aspects may help to foster the implementation of wearable sensors technology into rehabilitation settings.

### 4.4. Training Effectiveness

The results from our meta-analysis should be carefully interpreted because the majority of the studies present an “unclear” risk of bias in most of the established domains. Even if higher than in previous works, the number of studies included is probably not enough to determine a more conclusive statement about the effectiveness of WDBR. Further, due to the high variabilities in terms of type and configuration of sensors, outcome measures, and biofeedback, we performed the overall analysis on multiple pathologies limiting the possibility to apply the results on a specific population; only one subgroup analysis on stroke patients has been performed. 

However, our meta-analysis showed possible positive effects on balance with add-on WDBR for PwND compared to a control treatment, where subjects performed the same activities with the only difference being the use of wearable devices. In specific, a higher effect on balance was found in dynamic postural stability (e.g., TUG) outcomes. Similarly, a clinically significant improvement in functional standing balance (e.g., BBS) was noted after intervention and maintained at follow-up in Parkinson and stroke population. It is noteworthy that improvements from baseline in the WDBR group seem to reach the minimally clinically important change (MCIC) in the BBS outcome, highlighting that WDBR could be a promising approach to stimulate clinically significant improvements in PwND. According to Tomlinson et al. [9], a five-point improvement in BBS is needed to reach the MCIC for people with Parkinson’s disease, while a six-point difference represents the amount of change needed to conclude that a “true” clinical change in balance has occurred in stroke population [63].

Even if considerations about MCIC are clinically relevant, this evidence should be taken carefully and verified in future RCT to confirm the suggested effects. 

Subgroup analysis has only been provided for four studies sharing similar sensor type (pressure sensors), biofeedback components (auditory signals with constant frequency related to performance), and population (stroke) [46,48,52,57]. This analysis suggested that the add-on WDBR provides statistically significant effects in gait speed compared to controls. Moreover, all four studies reported an improvement in gait speed higher than 0.1 m/s corresponding to the MCIC in stroke population.

Conversely, subgroups analysis on stroke population suggested no effects on TUG, probably because in these studies wearable devices were predominantly used during gait activities to increase active weight-bearing on the paretic feet, while the TUG test involves more complex activities such as posture transitions and turning. 

### 4.5. Suggestions for Design Rules and Implementation for Clinical Practice

Based on the findings reported in this review, we have reported some highlights and suggestions to design effective and user-friendly wearable devices, facilitating their adoption in everyday clinical practice.

Firstly, we found that motor-learning principles were not properly integrated with the wearable devices used to provide biofeedback and were not properly described in most of the treatments’ protocols. 

This is a major concern, as wearable devices should be capable of modulating the biofeedback provided according to motor-learning principles since we considered rehabilitation as a learning environment. Thus, new systems should be designed to provide constant and real-time biofeedback in the first phase of rehabilitation in which repetitive cognitive stimulation is relevant for motor learning, and a fading or reduced biofeedback in the latter phases of rehabilitation in which increased variability of the feedback is required to stimulate motor learning from the associative to the autonomous phase [15,64]. Following these principles, the possibility to regulate biofeedback threshold and intensity according to patient’s need and patient’s rehabilitation phase should be implemented.

Secondly, the variety in wearable devices (in terms of sensor types and configurations, and biofeedback components) makes it difficult to compare them in terms of effectiveness in biofeedback rehabilitation in specific patients’ populations.

As a consequence of this high variability, currently, there is no evidence to claim if one configuration of sensors is superior to another. In our opinion, the configuration of sensors, in terms of number and positioning, should take into account the expected outcome, patient comfort, and clinical practicality (time spent to set up and reproducibility) [65]. In particular, wearable sensors’ configurations measuring biomechanical parameters should be tailored to the activity being examined (e.g., standing, turning, and walking) taking into consideration pathology specific impairments (e.g., specific abnormalities of gait phases) to provide an effective biofeedback. 

Thirdly, the heterogeneity of the clinical and instrumental outcome measures for balance and gait. Often, the instrumental measurements provided by wearable devices were not comparable. To solve these problems, wearable devices should provide consistent, reliable, and reproducible outcome parameters (e.g., step length, cadence, and single and double support time to assess gait) that clinical trials should report to enable a proper comparison between studies. Moreover, new easy-to-use instrumental indexes should be implemented to get a more objective and detailed assessment. Instrumental indexes should be complementary to the clinical assessment since they are able to detect subtle alterations not always visible from the clinical score to give a more complete portrait, and, consequently, to help clinicians in defining tailored rehabilitation treatments.

Finally, some suggestions are needed to improve the quality of studies on this topic. Due to the lack of evidence, there is a need for comparative studies to define what the best type of feedback and/or sensor configuration is most useful for clinical practice. Caution should be taken when considering the results of the effectiveness of using different types of sensor configuration and biofeedback on different neurological diseases. Future studies should address these issues by implementing higher-quality randomized clinical trials with larger sample sizes to improve the generalizability of the results. 

## 5. Conclusions

Our systematic review provides a comprehensive picture of the use of wearable sensors in clinical practice for PwND. Biofeedback has mostly been provided in real-time during movement execution, using biomechanical sensors and positive reinforcement through auditory or visual modes. Pressure sensors and EMG were mainly used to improve weight-bearing and muscles recruitment on the paretic leg in stroke patients, while inertial sensors were used to control axial segments and limbs in Parkinson’s disease. Nevertheless, the best sensor configuration in terms of number and positioning of sensors is not yet clear. 

Due to the current quality of the literature, it was not possible to achieve any firm conclusions about the effectiveness of WDBR from this study. Add-on WDBR seems to provide possible positive effects on dynamic balance for people with neurological diseases.

Specific design rules on motor-learning principles, feedback components, sensor configuration, and clinical practicality should be integrated to improve the effectiveness of wearable devices biofeedback rehabilitation. Higher-quality randomized control trials with a larger sample size are needed to draw any reliable conclusion.

## Figures and Tables

**Figure 1 sensors-21-03444-f001:**
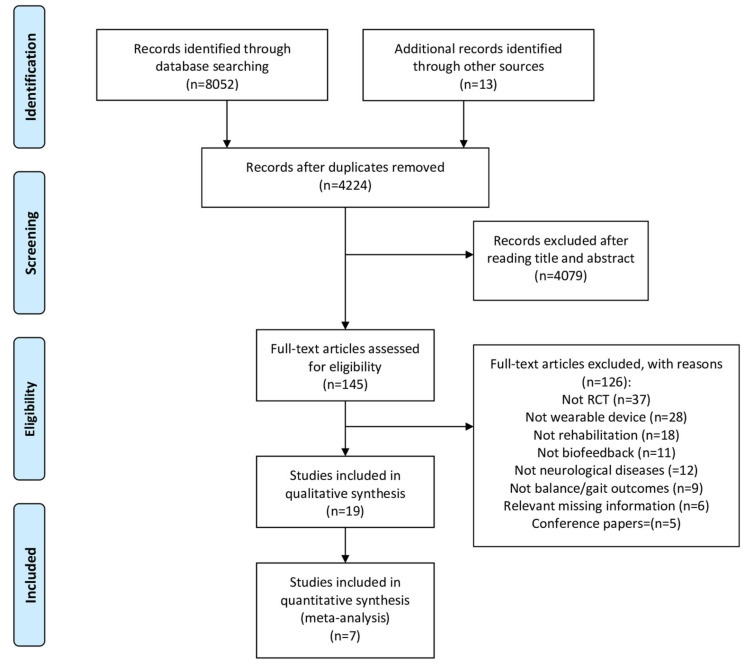
Flow diagram based on PRISMA statement (www.prisma-statement.org, accessed on 1 March 2020).

**Figure 2 sensors-21-03444-f002:**
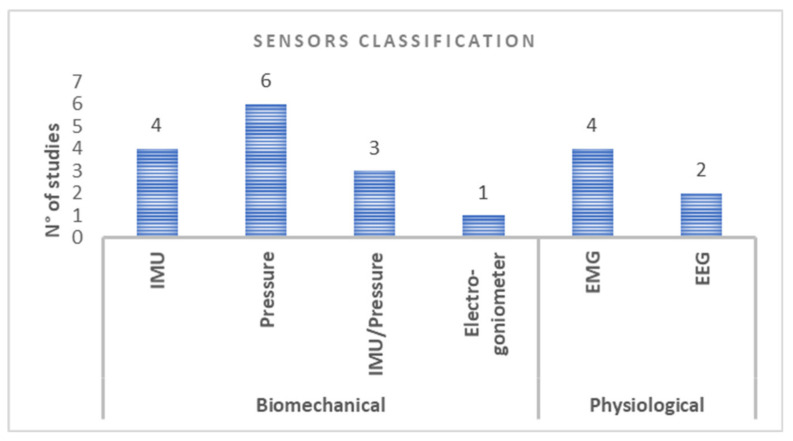
Classification of sensors used in the included studies: biomechanical (IMU, pressure and electrogoniometer) and physiological (EMG and EEG).

**Figure 3 sensors-21-03444-f003:**
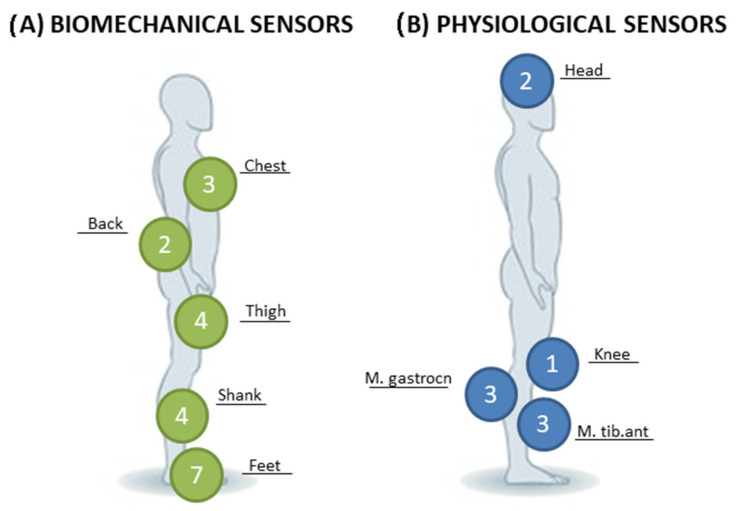
Sensors placement and number of the studies. (**A**) Biomechanical sensors: Chest: Carpinella et al., Lupo et al., and Van den Heuvel et al.; Back: Carpinella et al., Schwenk et al.; Thigh: Byl et al., Carpinella et al., Schwenk et al., and Lupo et al.; Shank: Byl et al., Carpinella et al., Lupo et al., and Schwenk et al.; Feet: Byl et al., Choi et al., Cha et al., El-tamawy et al., Ginis et al., Ki et al., and Sugkarat et al.; (**B**) Physiological sensors: Head: Azerpaikan et al. and Lee et al.; Knee: Cho et al.; M. Tibialis anterior: Cozean et al., Intiso et al., and Mandel et al.; M. Gastrocnemius: Jonsdottir et al., Mandel et al., and Cozean et al.

**Figure 4 sensors-21-03444-f004:**
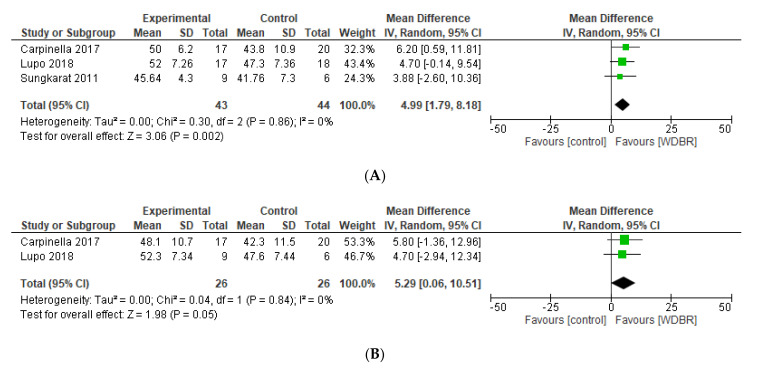
Forest plots of studies comparing add-on WDBR versus control. (**A**) Overall effect of WDBR versus control at postassessment on BBS. (**B**) Overall effect of WDBR versus control at follow-up assessment on BBS.

**Figure 5 sensors-21-03444-f005:**
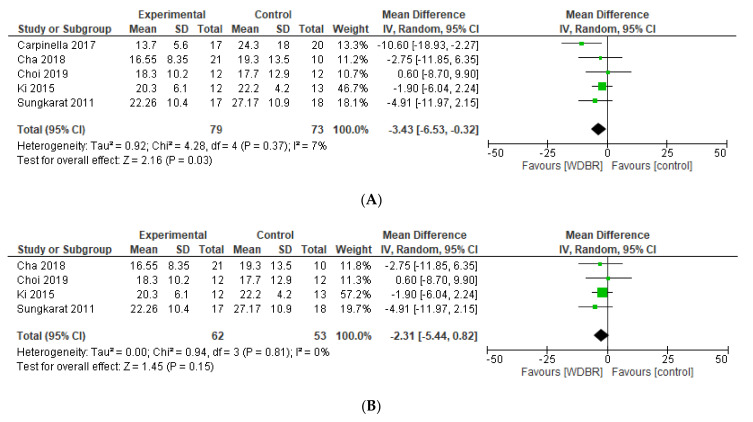
Forest plots of studies comparing add-on WDBR versus control. (**A**) Overall effect of WDBR versus control at postassessment on TUG. (**B**) Subgroup analysis: effect of WDBR (pressure sensors with auditory biofeedback) versus control at postassessment on TUG.

**Figure 6 sensors-21-03444-f006:**
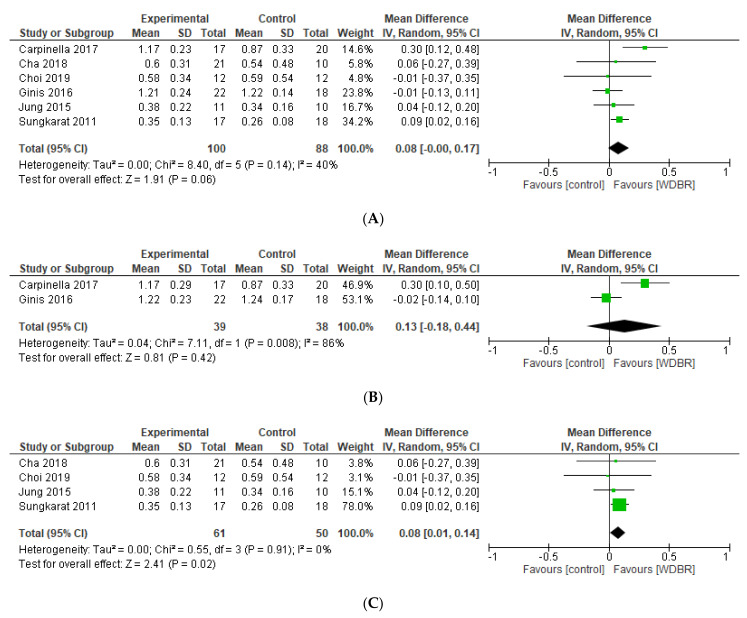
Forest plots of studies comparing add-on WDBR versus control. (**A**) Overall effect of WDBR versus control at postassessment on gait speed. (**B**) Overall effect of WDBR versus control at follow-up assessment on gait speed. (**C**) Subgroup analysis: effect of WDBR (pressure sensors with auditory biofeedback) versus control at postassessment on gait speed.

**Figure 7 sensors-21-03444-f007:**
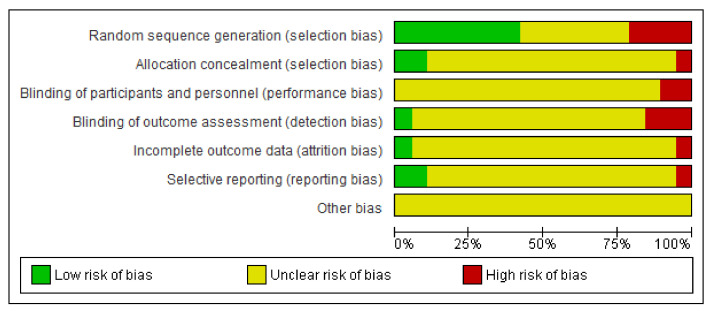
Risk of bias graph: review authors judgments about each risk of bias item presented as percentages across all included studies.

**Figure 8 sensors-21-03444-f008:**
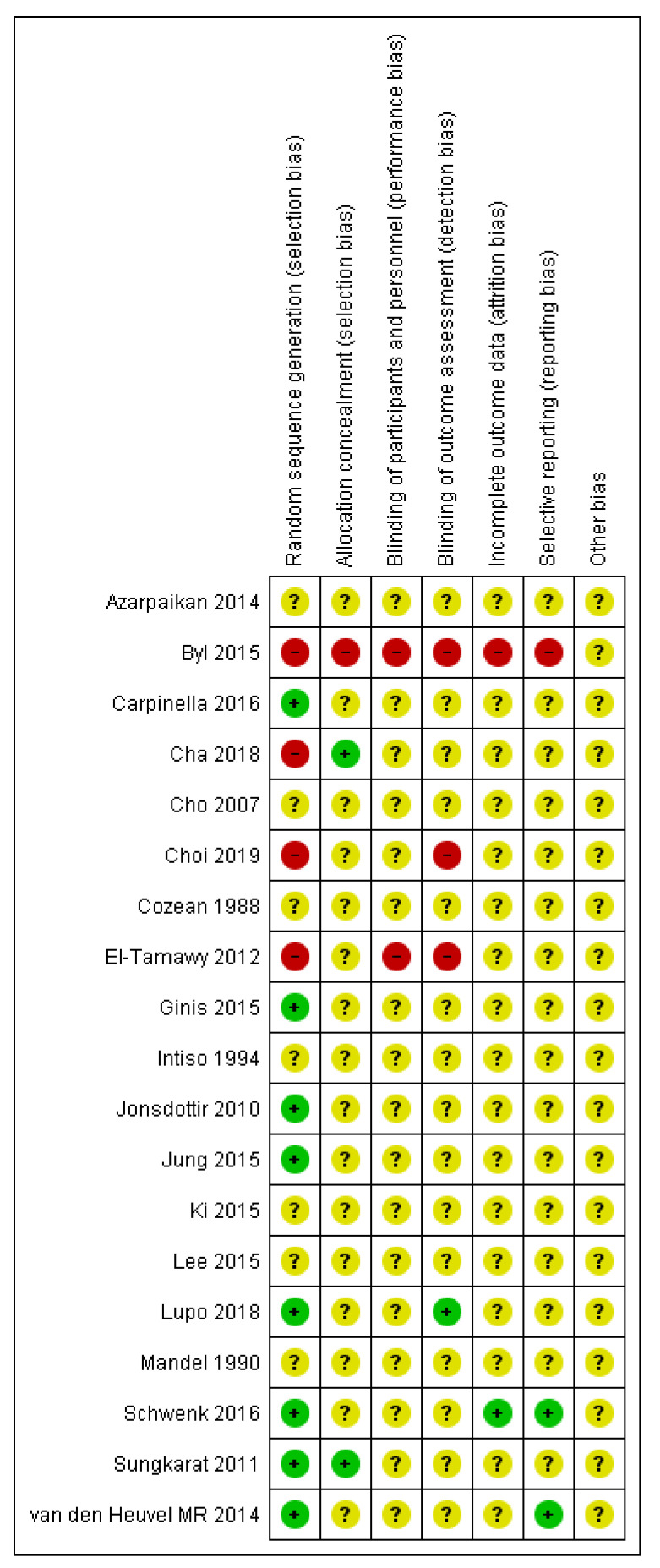
Risk of bias summary: review authors’ judgements about each risk of bias item for each included study.

**Table 1 sensors-21-03444-t001:** Search strategy example on Pubmed.

AND	(Humans[mh] OR Adult[mh] OR Nervous System Diseases[mh] OR Gait Disorders, Neurologic[mh] AND (Neurofeedback[mh] OR Feedback, Sensory[mh] OR feedback[tiab] OR Biofeedback[tiab] OR Cues[mh] OR Physical Therapy modalities[mh] OR Rehabilitation[mh] OR Rehab*[tiab] OR Conservative treatment[mh] OR Training[tiab] OR Exercise*[tiab])
	(Wearable Electronic Devices[mh] OR wearable[tiab] OR Device[tiab] OR Accelerometry[mh] OR Acceleromet*[tiab] OR gyroscope*[tiab] OR sensor*[tiab] OR shoe*[tiab] OR Insole*[tiab])
	(Walking[mh] OR Walk*[tiab] OR Ambulation[tiab] OR Gait[mh] OR Gait[tiab] OR Postural Balance[mh] OR Balance[tiab] OR Equilibrium[tiab] OR Recovery of function[mh] OR Motor Activity[mh])
	(Randomized controlled trial[pt] OR randomized controlled trials as topic[mh] OR random*[tiab])

**Table 2 sensors-21-03444-t002:** Characteristics of included studies.

First Author [Ref]	Aim	Participant Characteristics	Wearable Device	Sensors Type (Short)	Experimental Intervention	Control Intervention	Timing	Balance and Gait Outcome Measure	Evaluation Time Points
Azerpaikan et al. [40]	To study the effect of a neurofeedback training on balance problems associated with Parkinson’s disease.	TOT n = 16 patients with PD EG: n = 8 Mean age: 74.23 ± 3.51 years Female: n = 4 CG: n = 8 Mean age: 75.16 ± 3.64 years Female: n = 4	Biograph Infiniti Software system (version 5.0), the ProComp differential amplifier (Thought Technology Ltd, Montreal, Quebec) for NeuroFeedback training sessions (FlexComp Infiniti encoder, TT-USB interface unit, fiber optic cable, USB cable)	EEG SENSORS	EG: Neurofeedback training with EEG generator	EG: Sham Neuro Feedback Training using sham EEG generator	D:8 sessions T: 30 min F: 3 days/week	BBS*, Limit of stability *	PRE, POST
Byl et al. [41]	To evaluate the effectiveness of supervised gait training with and without visual kinematic feedback	TOT: n = 24 stroke and PD EG: n = 12 Stroke: n = 5 Mean age: 66.2 ± 5.0 years Female: n = 3 PD: n = 7 Mean age: 68.5 ± 3.6 years Female: n = 4 CG: n = 12 Stroke: n = 7 Mean age: 60.8 ± 5.4 years Female: n = 5PD: n = 5 Mean age: 70.0 ± 2.9 years Female: n = 2	Wireless joint angle sensors and Smart Shoes	IMU, PRESSURE SENSORS	EG: visual kinematic feedback on the computer screen during progressive and task-oriented balance and gait training activities	CG: balance and gait training activities	D: 12 sessions (6–8 weeks) T: 90 min (30 min visual kinematic feedback in the EG) F: NA	Gait Speed; Step Length; TBS, 6MWT, DGI, 5TSS, TUG, BBS, FOG-Q	PRE, POST
Carpinella et al. [42]	To test the feasibility of a wearable biofeedback system in a typical rehabilitation gym and analyze the effect on balance and gait outcome measures compared to physiotherapy without feedback.	TOT: n = 42 subjects with PD EG: n = 17 Mean Age: 73 ± 7.1 years Female: n = 3 CG: n = 20 Mean Age: 75.6 ± 8.2 years Female: n = 11	Gamepad System (IMU, PC and Customized software)	IMU SENSORS	EG: balance and gait functional tailored exercises using Gamepad System	CG: personalized balance and gait exercises defined by the clinical staff	D: 20sessions T:45 min F:3 times/week	BBS *, Gait Speed, UPDRSIII, TUG, ABC, FOGQ, COP ML sway *, COP AP sway, Tele-healthcare Satisfaction Questionnaire-Wearable Technologies	PRE, POST, FU
Cha et al. [46]	To compare the effectiveness of auditory feedback stimulation from the heel and forefoot areas in terms of ambulatory functional improvements in stroke patients	TOT n = 31 stroke subjects EG1: n = 11 Mean age: 64.6 ± 10.6 Female: n = 3 EG2: n = 10 Mean age: 63.0 ± 4.7 Female: n = 3 CG: n = 10 Mean age: 61.8 ± 9.8 Female: n = 4	A PedAlert Monitor 120 (PattersonMedical Holdings Inc., Warrenville, IL, USA)	PRESSURE SENSORS	EG1: gait training with active weight bearing on the paretic heel with auditory feedback EG2: gait training with auditory feedback from paretic metatarsals	CG= gait intervention	D: 6 weeks T: 50 min (30 min of conventional therapy + 20 min of gait intervention with or without auditory feedback) F: 3 × week	Gait speed, FGA *, TUG, COP length EO *, COP length EC, COP velocity EO *, COP velocity EC	PRE, POST
Cho et al. [47]	To examine whether visual biofeedback tracking training can improve gait performance in chronic stroke patients.	TOT n = 10 Stroke subjects EG: n = 5 Mean age: 46.2 ± 7.3 years Female: n = 2 CG: n = 5 Mean age: 48.8 ± 6.3 years Female: n = 1	A double-axis electrogoniometer (Biometrics Ltd. Ladysmith, VA) was used to record the instant degrees of knee joint flexion–extension. Series of PC generated sine waves at 0.2 Hz were displayed on a PC monitor at 80 cm distance from the eyes of the subject	ELECTRO-GONIOMETER	EG: visual biofeedback tracking training	CG: not Reported	D:20 sessions (4 week) T: 39 min F: 5 days/week	Motoricity Index, Modified Motor Assessment Scale, Gait Speed.	PRE, POST
Choi et al. [48]	To compare gait intervention with auditory feedback induced by active weight bearing on the paralyzed side with the effects of the general gait training method	TOT n = 24 stroke subjects EG: n = 12 Mean age: 62.8 ± 4.8 Female: n = 4 CG: n = 12 Mean age: 59.7 ± 10.2 Female: n = 4	PedAlert Monitor 120, (Patterson Medical Holdings, Inc.).	PRESSURE SENSORS	EG: gait intervention with auditory feedback	CG: general gait training over the ground.	D: 6 weeks T: 50 min (30 min of conventional therapy + 20 mins of gait training with or without bfb) F: 3 × week	Gait Speed (10MWT) *, FGA *, TUG *, COP length EO *, COP length EC*	PRE, POST
Cozean et al. [49]	To study the effect of EMG Biofeedback and FES as therapies for gait dysfunction in patients with hemiplegia after stroke.	TOT n = 36 Patients with stroke EG1: n = 9 Mean age:51 ± (not specified) Female: n = 4 EG2: n = 10 Mean age: 52 ± (not specified) Female: n = 2 G3: n = 8 Mean age: 56 ± (not specified) Female: n = 6 CG: n = 9 Mean age: 62 ± (not specified) Female: n = 2	Not reported	EMG SENSORS	EG1: EMG biofeedback during static and dynamic activities EG2: FES during static and dynamic activities EG3: EMG biofeedback+FES during static and dynamic activities	CG: conventional Physical Therapy	D: 6 week T: 30 min F: 3 × Week	ankle angle (swing phase) *, knee angle (swing phase) *, stride length, gait speed*	PRE, POST
El-Tamawy et al. [43]	To examine the influence of paired proprioceptive cues on gait parameters of individuals with PD.	TOT n = 30 subjects with PD EG: n = 15 Mean age: 61.4 ± 7.28 Female: not specified CG: n = 15 Mean age: 63.2 ± 5.6 Female: not specified	The vibratory device, OPTEC Co. LLtd.	PRESSURE SENSORS	EG: individually designed physiotherapy and traditional gait training plus treadmill training with vibratory stimuli	CG: individually designed physiotherapy and traditional gait training including instructions to walk with long steps.	EG: D: 8 weeks T: 51–70 min F: 3 sess/week, CG: D: 8 weeks T: 45 min F: 3 sess/week	Cadence *, Stride length *, Gait speed *, Walking distance *	PRE, POST
Ginis et al. [44]	To test the feasibility of CuPID system in the home environment and verify differential effects of CuPID training versus conventional home-based gait intervention	TOT n = 40 subjects with PD EG: n = 22 Mean age: not specified Female: not specified CG: n = 18 Mean age: not specified Female: not specified	The CuPiD system consisted of a smartphone (Galaxy S3-mini, Samsung, South Korea), a docking station and two IMUs (EXLs3, EXEL srl., Italy)	IMU SENSORS	EG: received weekly home visit and patients were instructed to walk with the CUPID system	CG: received weekly home visit by the researcher who gave advice on gait and freezing, and patients were instructed to walk without using the CUPID system	D: 6 weeks T: 30 min F:3 times/week	MiniBEST *, FSST, FES-I, 2MWT, UPDRS III, NFOG-Q, Comfortable gait and Dual task activities (gait speed, stride length, DS time);	PRE, POST, FU
Intiso et al. [50]	to evaluate the efficacy of electromyographic biofeedback compared with physical therapy.	TOT n = 16 Patients with stroke EG: n = 8 Mean age: 61.3 ± 12.3 years Female: n = 4CG: n = 8 Mean age: 53.5 ± 18.5 years Female: n = 3	table Satem PT 1015 and a walking Satem EMG Combitrainer PT 9115.	EMG SENSORS	EG: EMG Biofeedback and Physical Therapy (standard exercise bobath, facilitation, and inhibition techniques, neurofacilitatory techniques)	CG: Physical therapy (standard exercise bobath, facilitation and inhibition techniques, neurofacilitatory techniques)	D: 2 months T: 60 min F: daily physical therapy (Only EG 30 session of EMG BFB)	Basmajian scale *, Gait speed, step length, ankle angle (swing phase) *, ankle angle (heel contact)	PRE, POST
Jonsdottir et al. [51]	to assess the efficacy of EMG-BFB applied in a task-oriented approach based on principles of motor learning to increase peak ankle power of the affected leg and gait velocity in patients with chronic mild to moderate hemiparesis	TOT n = 20 Patients with stroke EG: n = 10 Mean age: 61.6 ± 13.1 years Female: not specified CG: n = 10 Mean age: 62.6 ± 9.5 years Female: not specified	BFB device: (SATEM Mygotron, SATEM srl, Rome, Italy); EMG, system (band-pass filtered at 20 to 950 Hz and then amplified with a gain of 40 000 (50 mVrms range),	EMG SENSORS	EG=Task-oriented gait training with EMG BFB device	CG= conventional physical therapy (at least 15 mins of gait training in each session)	D: 20 session T: 45 min F: 3 × Week	Gait speed *, ankle power peak at push-off *, stride length *, knee flexion peak	PRE, POST, FU
Jung et al. [52]	to examine the effect of gait training using a cane with an augmented pressure sensor to improve weight bearing on the nonparetic leg in patients with stroke	TOT 22 stroke subjects EG: n = 12 Mean age: 56.4 ± 11.1 Female: n = 4 CG: n = 10 Mean age: 56.3 ± 17.1 Female: n = 3	An instrumented cane, outfitted with a pressure sensor (CD 210-K200, Dacell Co. Ltd., Korea) connected to an indicator (DN30W, Dacell Co. Ltd., Korea)	PRESSURE SENSORS	EG: gait training with auditory feedback	CG: gait training without auditory feedback	D: 4 weeks T: 60 min (30 min +30 min gait training with or without bfb) F: 5 × week	Peak force cane *, Gait speed *, single support time *	PRE, POST
Ki et al. [53]	to examine the effects of auditory feedback during gait on the weight bearing of patients with hemiplegia resulting from a stroke.	TOT n = 30 stroke subjects EG: n = 12 Mean age: 55.3 ± 9.2 Female: n = 4 CG: n = 13 Mean age: 60.1 ± 12.3 Female: n = 2	A pressure gauge Ped-AlertTM120 (ORBITEC, USA)	PRESSURE SENSORS	EG: neurodevelopmental treatment with auditory feedback	CG: neurodevelopmental treatment	D: 4 weeks T: NA F: NA.	TUG *, Stance phase duration, Single support time.	PRE, POST
Lee et al [54]	to examine the effect of neurofeedack training on brain waves control and gait performed under dual-task conditions.	TOT n = 20 stroke subjects EG: n = 10 Mean age: 53.2 ± 6.46 Female: n = 4 CG: n = 10 Mean age: 54.7 ± 3.77 Female: n = 3	The Procomp Infiniti system (SA7951 version 5.1, Thought Technology, Canada) was used for neurofeedback training. The QEEG-8 (LXE3208, LAXHA Inc., Korea) system was used to measure brain waves	EEG SENSORS	EG: neurofeedback training	CG: pseudo-neurofeedback training (sham neurofeedback)	D: 8 weeks T: 30 min F: 3 × week	gait speed *, cadence *, stance phase percentage, and plantar foot pressure (dual task) *	PRE, POST
Lupo et al. [55]	To evaluate the efficacy of training involving the use of a combined biofeedback system versus conventional balance training	TOT n = 15 stroke subjects EG: n = 9 Mean Age: 52.56 ± 13.92 Female: n = 3 CG: n = 6 Mean Age: 65.66 ± 9.64 Female: n = 1	The RIABLO™ (CoRehab, Trento, Italy) system comprised of several inertial measurement unitsand a force platform connected wirelessly to a computer	IMU SENSORS AND FORCE PLATFORM	EG: balance training with RIABLO biofeedback system using a video interface.	CG: conventional balance training without the use of the RIABLO biofeedback system	D: 10 sessions T: 20 min F:3 times/week	BBS *, RMI, COP length EO *, COP length EC *	PRE, POST, FU
Mandel et al. [56]	to investigate the efficacy of electromyographic (EMG) versus a novel biofeedback (BFB) approach to improve ankle control and functional gait in stroke patients	TOT n = 37 stroke subjects EG1: n = 13 Mean age: 54.7 ± 13.9 years Female: n = 5 EG2: n = 13 Mean age: 57.5 ± 14.2 years Female: n = 5 CG: n = 11 Mean age: 56.8 ± 12.8 years Female: n = 1	Two channels of EM-BFB, a Lamoureux-type parallelogram electrogoniometer was and a computerized system to provide audiovisual feedback of ankle position during dorsiflexion and plantar flexion.	EMG SENSORS and ELECTRO-GONIOMETER	EG1: EMG biofeedback training (computer-generated auditory and visual feedback of calf and pretibial muscle activity during active ankle movements) EG2: EMG biofeedback followed by rhythmic positional biofeedback (computer-generated single-channel feedback of dorsiflexion and plantar flexion)	CG: No training	D: 24 session (EG2 performed 12 session of EMG biofeedback and 12 session of Rythmical positional biofeedback) T: not specified F: 3 × Week	Gait speed *, ROM *	PRE, POST, FU
Schwenk et al. [58]	To evaluate the feasibility and experience in using the new sensor-based training in a sample of patients with clinically confirmed amnestic MCI	TOT n = 32 subjects with MCI EG: mean age 77.8 ± 6.9 Female: n = 7 CG: mean age 79 ± 10.4 Female: n = 5	The technology consisted of a 24 inch computer screen, an interactive virtual user interface, and 5 inertial sensors (LegSysTM, BioSensics LLC, MA, USA)	IMU SENSORS AND FORCE PLATFORM	EG: postural balance exercises during standing (ankle point-to-point reaching tasks and virtual obstacles crossing tasks) using biofeedback training	CG: No training	EG: D: 4 weeks T: 45 mins F: 2 sessions/week	CoM area EO *, CoM area EC, CoM sway ML EO *, CoM swayML EC, CoM sway AP EO *, CoM sway AP EC, Gait Speed, Gait stride time variability, FES-I *	PRE, POST
Sungkarat et al. [57]	To determine whether improved symmetrical weight bearing somatosensory feedback would result in improved gait and balance in people with stroke	TOT n = 35 people with stroke EG: n = 17 Mean Age: 52.12 ± 7.17 years Female: n = 5 CG: n = 18 Mean Age: 53.83 ± 11.18 years Female: n = 6	Insole Shoe Wedge and Sensors (I-ShoWS)	PRESSURE SENSORS	EG: conventional rehabilitation and gait training with I-ShoWS set-up	CG: conventional rehabilitation and gait training without I-ShoWS set-up	D:15 sessions T: 60 min/session, (30 min gait training and 30 min conventional) F: 5 days/week	Gait speed *, Step Length asymmetry ratio*, Single Support time asymmetry ratio *, Load on paretic leg during stance (% Body Weight) *, BBS *, TUG *	PRE, POST
van den Heuvel et al. [45]	to investigate the feasibility of visual feedback-based balance training (VFT) and to compare the effects of the training program with conventional training	TOT n = 33 subjects with PD EG: mean age 63.9 ± 6.39 Female: n = 5 CG:mean age 68.8 ± 9.68 Female: n = 8	Flat-panel LCD monitor connected to a PC (Motek Medical, Amsterdam, The Netherlands), Force plate (Forcelink, Culemborg, and The Netherlands) and Inertial sensors (X sens, Enschede, The Netherlands)	IMU SENSORS	EG: interactive balance games with explicit augmented visual feedback	CG: conventional balance training recommended by the guidelines for physical therapy.	D: 5 weeks T: 60 min (45 min balance workstation) F: 2 sess/week	BBS; Single leg stance test; Gait speed, FES-I, UPDRSIII*	PRE, POST, FU

* = *p* ≤ 0.005; 5TSS = 5 Time sit to stand; 2MWT = 2 min Walk Test; 6MWT = 6 min Walk Test; AP = Antero-posterior; BFB = Biofeedback; BBS = Berg balance scale; CG = Control group; COM = Centre of movement; COP = Centre of pressure; D = Duration; DGI = Dynamic Gait Index; EEG = Electroencephalogram; EG = Experimental group; EMG = Electromyography, F = Frequency; FES-I = Fall efficacy scale; FGA = Functional gait assessment; FOG-Q = Freezing of gait questionnaire; FSST = Four Square step test; IMU = Inertial movement unit; MCI = Mild cognitive impairment; ML = Medio-lateral; NFOG-Q = New freezing of gait questionnaire; PD = Parkinson disease; RMI = Rivermead mobility index; ROM = Range of motion; T = time; TBS = Tinetti balance scale; TOT = Total; TUG= Timed up and go; UPDRS = Unified Parkinson disease rating scale.

**Table 3 sensors-21-03444-t003:** Biofeedback components.

Author, Year	Biofeedback Mode	Biofeedback Content	Biofeedback Frequency	Biofeedback Timing	Reinforce Type
Azerpaikan, 2014	Visual	Performance	Constant	Concurrent	Positive
Byl, 2015	Visual	Performance, Result	Constant	Terminal	Positive, Negative
Carpinella, 2017	Auditory, Visual	Performance, Result	Fading	Concurrent, Terminal	Positive, Negative
Cha, 2018	Auditory	Performance	Constant	Concurrent	Positive
Cho, 2007	Visual	Performance	Constant	Concurrent	Not specified
Choi, 2019	Auditory	Performance	Constant	Concurrent	Positive
Cozean, 1988	Auditory, Visual	Performance	Constant	Concurrent	Positive
El-Tamawy, 2012	Vibrotactile	Performance	Constant	Concurrent	Positive
Ginis, 2016	Auditory	Performance	Fading	Concurrent	Positive, Negative
Intiso, 1994	Auditory	Performance	Constant	Concurrent	Positive
Jonsdottir, 2010	Auditory	Performance	Fading	Concurrent	Positive
Jung, 2015	Auditory	Performance	Constant	Concurrent	Negative
Ki, 2015	Auditory	Performance	Constant	Concurrent	Positive
Lee, 2015	Visual	Performance	Constant	Concurrent	Positive
Lupo, 2018	Auditory, Visual	Performance, Result	Constant	Concurrent	Not specified
Mandel, 1990	Auditory, Visual	Performance	Constant	Concurrent	Positive
Schwenk, 2016	Auditory, Visual	Performance	Constant	Concurrent	Positive, Negative
Sungkarat, 2011	Auditory	Performance	Constant	Concurrent	Positive
van den Heuvel, 2014	Visual	Performance, Result	Constant	Concurrent	Positive

## Data Availability

Not applicable.

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
