# Peer review of "Wearable Devices for Biofeedback Rehabilitation: A Systematic Review and Meta-Analysis to Design Application Rules and Estimate the Effectiveness on Balance and Gait Outcomes in Neurological Diseases"

_sensors, 2021, doi:10.3390/s21103444_

Round 1
Reviewer 1 Report
Please see the attached document for major and minor comments.

Author Response
Please, find attached the reply

Reviewer 2 Report
Refer to attachment.

Author Response
please, find attached the reply

Reviewer 3 Report
The manuscript contains a systematic review on wearable devices for biofeedback in balance and gait rehabilitation for neurological diseases. The methodology appears to be very cleanly executed and the results were systematically presented for the most part. What is missing, on the one hand, is an introduction that clearly and specifically formulates and motivates the topic and the challenges in the application of wearable devices for biofeedback in balance and gait rehabilitation for neurological diseases. On the other hand, there is too little critical questioning in the discussion and conclusions are not drawn from the own review results but from other sources. Therefore, I recommend a comprehensive revision of the introduction and discussion.
Introduction:
The review focuses on technology-assisted biofeedback (BF) of balance and gait rehabilitation in neurological diseases. This selection has to be better introduced in the beginning of the manuscript. Please introduce what patient groups / diseases are meant in particular, what kind of gait & balance disturbances occur and why this aspect of reha is so important (relevance, challenges), and what are the differences in PD and stroke regarding the rehabilitation goals.
The transition from BF provided by clinicians to BF wearables seems very abrupt - an introduction why wearables are useful and can relieve clinicians and help patients (e.g., home training, tele-medicin).
Imho, the term (a) „inertial measurement unit“ is more common than (b) „inertial movement unit“ for the abbreviation IMU. The authors use (b) in the text, but (a) in the keywords, which is inconsistent. Also, it might have been good to include this term in the search terms (Tab. 1).
The paragraph from lines 78-84 is critical in my opinion, as it is very vague and not specific, and needs to be revised. What exactly is the best sensor combination being sought for, for which rehabilitation setting (e.g. clinic or home), for which therapy goal? Depending on the patient group, this can be very different. This should be taken into account when setting the goals here.
Methods:
Well written and state-of-the-art.
Please comment on the suitability of your keywords for the search as they made you find more than 4000 articles where in the end, you were only able to use 20 of them.
Results:
Table 2: It is a great challenge to summarize the information of 20 studies in one table and the authors have only partially succeeded. At first glance, the table does not seem so clear, but together with Table 3, some things become clearer. My recommendation would be to highlight the following points in table: disease, number of participants, sensor type and biofeedback mode. This can be achieved, for example, by a better, clearer formatting (participant characteristics) and by reducing the information to the necessary minimum (e.g., wearable device and sensor placemnt are often quite wordy). Besides, it is not entirely clear to me what criteria were used to sort the literature. By year of publication? Then it would be good if this also appears in column 1. Also, please also comment on why the study of Aruin et al. was included, as many important details on the methodology of the study are obviously missing (e.g., group size of CG and EG, sensor type). Please consider removing this study from the selection.
In meta-analysis, the mix of patient groups appears difficult and should be discussed at least insistently later.
Discussion:
Be more precise (first paragraph): The authors present a review on BF for gait & balance rehabilitation only. The authors claim to present design rules but they have not presented them until this point in the manuscript; therefore, this sentence and the following belongs into the conclusions.
Training paradigm: This section is rather a summary than a discussion. Please refrain from repetition of the study details; for example, the details on the EEG studies are more precise than in the results section (move there?). What is missing is an evaluative comparison of the techniques at this point related to effort, handling, accuracy, …
Biofeedback components: Few conclusions were drawn from the new analysis and reference is rather made to known knowledge. Would it be possible to, for example, compare the outcome of the studies considered regarding positive and negative reinforcement?
Feasibility and Usability of Sensor-Based Training: What about effort? What is critical about the fact that usability was not evaluated in most of the studies?
Suggestions for design rules and implementation for clinical practice: „Our findings highlight that specific design rules should be integrated to improve the effectiveness of wearable devices biofeedback rehabilitation.“ This statement could not be inferred from the previous text. Please explain in more detail. Also the „rules“/“recommendation“ that are drawn in the second paragraph do seem to come directly from the results of the literature review. What exactly were the core problems identified in the strategies & studies and how can they be improved?
Abstract:
After the Introduction and Discussion have been revised, the Abstract should also be adapted accordingly. Currently, it seems very unfavorable to state that the review was conducted to answer the question about the effectiveness of WDFR, and then exactly that point cannot be answered.
Language, Grammar and Style:
Careful proof-reading is necessary regarding careless mistakes and consistency, e.g. missing spaces, missing sentence-dots, missing commas, missing articles. Here are some examples:
-
many missing spaces between text and text/ref in brackets (either „(„ or „[„“); e.g. abbrev. in abstract lines 16, 17, 22, 49
-
missing sentence dots: e.g. line 41, 49
-
line 41: Incorrect sentence start
-
missing dots in abbrev „e.g.“ (inconsistent)
-
define abbrev. PD once in the text
-
define abbrev. RCT once in the text (def. in abstract ist not sufficient)
-
Table 2, list of abbreviations: please sort them according to the alphabet, otherwise the search is very tedious for the reader
-
Table 2: Missing abbreviation definitions: TOT, D, T, F
-
Table 2: Time Points – definition in the text is different than the meaning in the table
-
Page numbers are incorrect
-
It would be nice if the ref-numbers of the corresponding articles could appear in the figures of the meta-analysis
Author Response
Please, find attache the reply

Round 2
Reviewer 2 Report
Please refer to attachment.

Author Response
ANSWER to MINOR REVISION
Thank you for your considerations. The abstract has been modified. In the Discussion, the “Feasibility and Usability of Sensor-Based Training” section showed a typo that has been corrected. Finally, a careful proof reading by and English editor has been provided before the publication.

Reviewer 3 Report
The manuscript has been revised with regards to the comments of both reviewers with particular improvements in introduction and discussion.
Abstract: The abstract seems unfinished with placeholders such as „…“, and the last sentence appears inconclusive. This has to be fixed before publishing!
Introduction:
The introduction has been revised and now motivates the use of wearables in rehabilitation in a better and more detailed way. However, the linguistic elaboration lacks care and the text could be further smoothed. Here are some examples regarding care:
L 41: „ke“ instead of stroke, Comma-dot combination?
L 46: PD patients ususally undergo „PD therapy“ rather than „PD rehabilitation“
L. 58, 60 ++: Space is missing „combination.Content“, „ [19].Frequency“, „process.Timing“, „allowingcontinuous“, „improvingrehabilitation“; this formatting problem runs through the whole document (e.g. Discussion section „onthe“, „tocontrol“, „Firstlywe“, „Secondlythe“)
Therefore, another careful proof-reading is necessary with focus on the revised parts.
Discussion:
L 619+: „Most of the studies failed to evaluate the feasibility and the usability: this is a major limit. This means that we have many clues from literature on strategies to reduce the effort required to use wearable devices by both therapists and patients, to reduce the time spent setting up, to regulate biofeedback threshold and to improve data extraction.“ These two sentences contradict each other in my opinion. It does not follow from the statement that „the studies failed to evaluate the feasibility and usability“ that „we have many clues“ on how to improve this aspects. It rather follows that there is a lack of systematic investigation, possibly although there are indications in the literature that this is needed. Please rephrase!
Language (Examples):
L. 719: Parkinsons Disease
L. 720: best sensor configuration
L. 723: consistency: Add-on WDBR
Author Response
ANSWER to REVIEWER 3
Thank you for your comments. We agree with your considerations.
Regarding PICO we defined "C" to be clear with the reader in terms of possible comparison. All the studies have a control group but in some studies the control group received "no training". It is true that we had missed "no training" as a possible comparison in PICO, so we have added it.
Considering the inclusion/exclusion criteria we have added a clearer section for the readers (LINE 134-143) after the PICO. In the inclusion criteria the ‘O’ of the PICO has been better defined.
Regarding the inclusion of only RCTs, in the Introduction section we have clearly stated that RCTs are needed as the most robust design to determine effectiveness of intervention. In this way our intention to include only RCTs should be clear for the reader. This has been confirmed in methods.
In the conclusion, as you suggested, we have added that our review it is comprehensive for people with neurological diseases.
Finally, a careful proof reading has been provided by an English editor before the publication.
